# Optimal Morphometric Characteristics of a Tubular Polymeric Scaffold to Promote Peripheral Nerve Regeneration: A Scoping Review

**DOI:** 10.3390/polym14030397

**Published:** 2022-01-20

**Authors:** Josefa Alarcón Apablaza, María Florencia Lezcano, Karina Godoy Sánchez, Gonzalo H. Oporto, Fernando José Dias

**Affiliations:** 1Research Centre in Dental Sciences (CICO-UFRO), Dental School—Facultad de Odontología, Universidad de La Frontera, Temuco 4780000, Chile; josefa.alarcon@ufrontera.cl (J.A.A.); florencia.lezcano@ufrontera.cl (M.F.L.); gonzalo.oporto@ufrontera.cl (G.H.O.); 2Program of Master in Dental Science, Dental School, Universidad de La Frontera, Temuco 4780000, Chile; 3Department of Integral Adults Dentistry, Dental School—Facultad de Odontología, Universidad de La Frontera, Temuco 4780000, Chile; 4Laboratorio de Cibernética, Departamento de Bioingeniería, Facultad de Ingeniería, Universidad Nacional de Entre Ríos, Oro Verde 3100, Argentina; 5Scientific and Technological Bioresource Nucleus (BIOREN-UFRO), Universidad de La Frontera, Temuco 4780000, Chile; karina.godoy@ufrontera.cl

**Keywords:** tissue engineering, nerve scaffold, morphology, peripheral nerve regeneration, regenerative biology

## Abstract

Cellular behavior in nerve regeneration is affected by the architecture of the polymeric nerve guide conduits (NGCs); therefore, design features of polymeric NGCs are critical for neural tissue engineering. Hence, the purpose of this scoping review is to summarize the adequate quantitative/morphometric parameters of the characteristics of NGC that provide a supportive environment for nerve regeneration, enhancing the understanding of a previous study. 394 studies were found, of which 29 studies were selected. The selected studies revealed four morphometric characteristics for promoting nerve regeneration: wall thickness, fiber size, pore size, and porosity. An NGC with a wall thickness between 250–400 μm and porosity of 60–80%, with a small pore on the inner surface and a large pore on the outer surface, significantly favored nerve regeneration; resulting in an increase in nutrient permeability, retention of neurotrophic factors, and optimal mechanical properties. On the other hand, the superiority of electrospun fibers is described; however, the size of the fiber is controversial in the literature, obtaining optimal results in the range of 300 nm to 30 µm. The incorporation of these optimal morphometric characteristics will encourage nerve regeneration and help reduce the number of experimental studies as it will provide the initial morphometric parameters for the preparation of an NGC.

## 1. Introduction

Peripheral nerve injuries have become a significant financial burden because their incidence has increased considerably in recent decades, bringing with them inherent morbidity, and lifelong disability [1,2]. Despite the nerve regeneration capacity of the peripheral nervous system (PNS), the anatomical recovery of damaged peripheral nerves, and sensory and motor functions after injury are not ideal [3,4,5]. The full section of the peripheral nerve is the most severe form of damage, especially when there is a large defect or “gap” [6,7,8,9]. The autograft is considered the gold standard treatment for this type of nerve injury, but it has several drawbacks, including sensory deficits caused by donor nerve surgery, limited nerve sources, neuroma formation, scarring, difficulties in repairing long segments, and the possibility of the nerve fibers not matching the donor nerve [4,5,8,9,10,11,12,13]. This is why knowledge of new techniques is essential to achieve successful regeneration [9,10].

Tissue-engineered “nerve guide conduits” (NGC) are an alternative to address the problems that accompany autograft and allograft-based nerve regeneration techniques [9,11,14,15]. The optimal characteristics of an NGC depend on biological, mechanical, and physical parameters [16]. Biological factors involve the selection of biocompatible and biodegradable materials that guide cell growth towards the development of three-dimensional tissue [5]. The mechanical properties ensure the stability of the scaffold materials, and the physical factors are determined by the external morphological characteristics of the scaffold, which involves the micro/macrostructure [17].

These NGCs are made from biomaterials that include metals, ceramics, and polymers [18]. Among these, polymers are widely preferred as scaffolds for peripheral and central nerve regeneration in both in vitro and in vivo studies [19,20]. Polymers are of great interest in the field of nerve regeneration because they have biodegradable, non-toxic/non-inflammatory and mechanical properties similar to the tissue to be replaced; are highly porous, which promotes cell attachment and growth; have economical and simple manufacturing processes; and have a potential for chemical modification leading to increased interaction with normal tissue [21,22,23].

An ideal nerve conduit should be resistant, flexible, porous, biocompatible, biodegradable, neuroconductive, and have appropriate surface and mechanical properties to promote nerve regeneration [24]. Although these polymeric biomaterials promise to fulfill some of the above-stated criteria, they have some drawbacks which must be overcome to meet the specific tissue engineering applications. Polymer blending offers one of the most successful methods of developing a suitable scaffold with all the preferred properties for these applications [21]. Furthermore, the morphological characteristics of the polymeric scaffold are a defining area in nerve regeneration [21,24].

Therefore, the surface properties of the polymeric material must be designed with specific topographic cues to enhance cellular interaction with the biomaterial and provide the optimal environment for peripheral nerve regeneration [25]. Part I of the present study described seven structural characteristics of an NGC considered important for promoting nerve regeneration, of which three are qualitative characteristics—adjustment of NGC, pore distribution, and NGC fiber alignment—and four quantitative characteristics—wall thickness, porosity, pore size and NGC fiber thickness [24].

Among the qualitative characteristics, it has been described that the diameter or adjustment of NGC to the nerve in which it will be implanted is decisive in nerve regeneration as it determines the mechanical properties and can influence the quality of the nerve regeneration [26,27]. On the other hand, pore distribution determines the vascularization, permeability, exchange of nutrients and retention of neurotrophic factors, avoiding undesirable cellular infiltration and always considering the need to maintain biomechanical and biodegradable properties [4,12,13]. Finally, the orientation of the fibers is important to promote the regeneration of the peripheral nerve since it could alter cell differentiation, morphology, growth, proliferation, and migration [28,29,30,31,32].

The importance of quantitative characteristics in nerve regeneration has been described in the literature. Wall thickness, porosity, and pore size are determinants in the exchange of molecules, such as nutrients, growth factors, glucose, lysozyme also determine the mechanical properties of NGC [24]. On the other hand, in polymeric NGC prepared by electrospinning, the thickness of the polymeric fibers takes on importance in nerve regeneration [28,29], since it could influence the alignment, growth, and density of the neurite, as well as the direction, migration, proliferation, and adhesion of the Schwann cells (SCs) [24].

However, the optimal parameters of the quantitative morphological characteristics that ensure nerve regeneration were not established in the literature. Therefore, the objective of this review is to describe the current knowledge on the appropriate parameters of the polymeric tubular scaffold quantitative characteristics that have been shown to provide a supportive environment for cell survival and development, mimicking the extracellular matrix (ECM) and normal anatomy, synergistically promoting morphogenesis, differentiation, and homeostasis of nervous tissues [3,33]. The determination of these optimal characteristics should reduce the number of experimental studies (in vitro and in vivo) as this will provide the initial morphological parameters for creating the scaffold. This represents Part II of two parts of the scoping review.

## 2. Materials and Methods

### 2.1. Systematic Literature Search

In a previous article, seven main characteristics were described that a nerve scaffold must have and which are decisive for nerve regeneration [24]. Of the seven characteristics described, four are quantitative, i.e., they can be used to examine and compare data numerically. The four quantitative characteristics described in the previous study served as the basis for the development of this manuscript [24]. In the present article, a new literature search was carried out to describe the parameters of the four quantitative morphological characteristics of an NGC to promote nerve regeneration [24]: porosity, pore size, wall thickness, and fiber thickness. This scoping review was reported according to the Preferred Reporting Items for Systematic reviews and Meta-Analyses extension for Scoping Reviews (PRISMA-ScR) guidelines [34]. The revised databases were PubMed / MEDLINE, Scopus, Scielo, and Web of Science.

The search terms selected were: “peripheral nerve”, “polymer”, “scaffold”, “nanofibroses”, “fibers”, “tissue engineering”, “tissue guide”, “tissue scaffolds”, “morphology”, “characteristic”, “features”, “morphometry”, “thickness”, “pore”, “porosity”, “diameter”, “wall”, “alignment”, “regeneration”, “nervous regeneration”, “neural growth”. The keywords were combined with Boolean operators OR and AND. The search was carried out between April 2020 and August 2021. A manual search of the literature was carried out by reviewing the references in the articles found in the electronic databases.

The studies included in the previous article [24] that analyzed the morphological characteristics of porosity, pore size, wall thickness, and fiber thickness that described a numerical comparison of two or more parameters were analyzed for inclusion

The following search equations was used in PUBMED:

((((((((“Neurogenesis”[Mesh]) OR (neural growth)) OR (nervous regeneration)) OR (nerve regeneration)) AND ((Polymer*) OR (“Polymers”[Mesh]))) AND ((((Scaffold) OR (“Tissue engineering”)) OR (“Tissue guide”)) OR (“Tissue scaffolds”))) AND (((((((((alignment) OR (“wall thickness”)) OR (wall thickness)) OR (wall)) OR (diameter)) OR (Porosity)) OR (“Porosity”[Mesh])) OR (pore size)) OR (¨pore size¨))) AND ((((features) OR (characteristic)) OR (Morphology)) OR (morphometry))) AND ((((“Nerve Fibers, Myelinated”[Mesh] OR “Nerve Fibers, Unmyelinated”[Mesh]) OR (fibers)) OR (electrospun nanofibers)) OR (nanofibers))

neural growth: “neurogenesis”[MeSH Terms] OR “neurogenesis”[All Fields] OR (“neural”[All Fields] AND “growth”[All Fields]) OR “neural growth”[All Fields]

nervous: “anxiety”[MeSH Terms] OR “anxiety”[All Fields] OR “nervous”[All Fields]

regeneration: “regenerability”[All Fields] OR “regenerable”[All Fields] OR “regenerant”[All Fields] OR “regenerants”[All Fields] OR “regenerate”[All Fields] OR “regenerated”[All Fields] OR “regenerates”[All Fields] OR “regenerating”[All Fields] OR “regeneration”[MeSH Terms] OR “regeneration”[All Fields] OR “regenerations”[All Fields]

nerve regeneration: “nerve regeneration”[MeSH Terms] OR (“nerve”[All Fields] AND “regeneration”[All Fields]) OR “nerve regeneration”[All Fields]

scaffold: “scaffold”[All Fields] OR “scaffold’s”[All Fields] OR “scaffolded”[All Fields] OR “scaffolder”[All Fields] OR “scaffolders”[All Fields] OR “scaffolding”[All Fields] OR “scaffoldings”[All Fields] OR “scaffolds”[All Fields]

alignment: “align”[All Fields] OR “alignability”[All Fields] OR “alignable”[All Fields] OR “aligned”[All Fields] OR “alignment”[All Fields] OR “aligner”[All Fields] OR “aligner’s”[All Fields] OR “aligners”[All Fields] OR “aligning”[All Fields] OR “alignment”[All Fields] OR “alignments”[All Fields] OR “aligns”[All Fields]

thickness: “thick”[All Fields] OR “thickness”[All Fields] OR “thicknesses”[All Fields]

diameter: “diameter”[All Fields] OR “diameters”[All Fields]

porosity: “porosity”[MeSH Terms] OR “porosity”[All Fields] OR “porosities”[All Fields]

features: “feature’s”[All Fields] OR “featured”[All Fields] OR “features”[All Fields] OR “featuring”[All Fields] OR “feature”[All Fields]

characteristic: “characteristic”[All Fields] OR “characteristics”[All Fields]

morphology: “anatomy and histology”[Subheading] OR (“anatomy”[All Fields] AND “histology”[All Fields]) OR “anatomy and histology”[All Fields] OR “morphology”[All Fields] OR “morphologies”[All Fields]

morphometry: “morphometries”[All Fields] OR “morphometry”[All Fields]

fibers: “fiber”[All Fields] OR “fibre”[All Fields] OR “fiber’s”[All Fields] OR “fiberized”[All Fields] OR “fibers”[All Fields] OR “fibre’s”[All Fields] OR “fibres”[All Fields]

nanofibers: “nanofibers”[MeSH Terms] OR “nanofibers”[All Fields] OR “nanofiber”[All Fields]

The same search equations were adapted for the other search engines.

### 2.2. Eligibility Criteria

Only primary studies in vitro and on animals were included, that declared as a general objective the study of one or more morphometric characteristics of a tubular polymeric scaffold in the PNS. Full-text articles with no limits on the publication date, published in English or Spanish, were included for the analysis. We excluded secondary studies or those published as abstracts. Studies performed on the central nervous system (CNS) and studies that did not present a measurement comparison for the assessment of the characteristics were not included.

### 2.3. Article Selection and Data Extraction

Titles and abstracts of studies retrieved using the search strategy were screened independently by two review authors. Articles that fulfilled the eligibility criteria were analyzed in full text to confirm their selection. The following information was collected from the full-text articles comprising the final selection: authors, publication year, study design, the polymer used, morphometric characteristics, method to prepare the NGC, and quantitative value of the characteristics studied. The main result evaluated in the studies was the optimal parameters of the morphometric characteristics that promote peripheral nerve regeneration.

## 3. Results

### 3.1. Study Selection

The article search and selection process are summarized in Figure 1. The total number of articles found in the databases was 358 citations. Nine additional articles were included after the manual search, and 27 were selected from the previous review and met the eligibility criteria, totaling 394 studies, of which 103 were duplicates. After the initial reading by title and abstract, 159 articles were ruled out, of which 24 were literature reviews, 62 studies of biomaterials with no regenerative outcomes, 45 studies assessed regeneration of the CNS, and 28 did not describe the parameters of one or more morphometric characteristics.

After reading full-text articles (132 in total), 103 were excluded, of which four were literature reviews, 31 were CNS studies, 24 evaluated new scaffold building techniques, and 41 included in their text the morphological and morphometric characteristics to be studied, but there was no measurement comparison for their assessment and three did not evaluate nerve regeneration. Finally, in this review, 29 articles were included that corresponded to experimental studies in vitro or in vivo that met the previously defined criteria.

### 3.2. Characteristics of the Selected Studies

The present article describes the optimal value parameters of four morphometric characteristics that should be considered in the creation of an NGC since they are determinants for regenerative success. The characteristics of the NGC are listed in Table 1, Table 2 and Table 3.

The morphometric characteristics of a polymeric NGC are decisive in its mechanical properties and peripheral nerve regeneration. Thirty studies evaluated the optimal parameters of one or more morphometric characteristics of an NGC. Ten in vitro and/or in vivo studies evaluated the optimal parameter of the wall thickness of a polymeric NGC, [23,25,33,35,36,37,44,45,47,52] of which six studied it as the main characteristic and five as a secondary one. Wall thickness variability was a determinant in the exchange of molecules and growth factors between the internal and external environment. Furthermore, wall thicknesses are decisive in mechanical properties and nerve regeneration.

Eleven studies evaluated the thickness of the electrospun polymeric fibers [5,8,10,28,29,38,39,40,41,42], with ten being in vitro and one in vivo. These articles describe the importance of the thickness of these polymeric fibers, mainly in the influence of neurite extension, density, and alignment, as well as the differentiation, migration, proliferation, and adhesion of SCs.

Ten in vitro and/or in vivo studies evaluated the porosity of the NGC [12,13,35,36,44,45,46,48,50,52], and 11 in vitro and/or in vivo studies evaluated the pore size [4,12,43,44,46,47,48,49,50,51,52]. These characteristics significantly influence the exchange of the nutrients, molecules, and growth factor necessary for nerve regeneration.

## 4. Discussion

### 4.1. Polymers

Ideal nerve tissue engineering scaffolds should be able to mimic the natural environment of the ECM, be biocompatible to promote cell interactions, have adequate mechanical properties during tissue regeneration, and be biodegradable, which avoids a second surgery after implantation [5,10,53]. A promising alternative is polymeric fibers that have been extensively studied due to their biocompatibility properties, controllable degradation rate, and flexibility. In addition, the morphology, hydrophilicity, surface energy, and charge of the scaffold control cell adhesion, migration, phenotypic maintenance, and intracellular signal transmission in vitro, are crucial factors that should be considered in the preparation of polymer fibers [53].

To date, different types of polymeric fibers have been used for tissue engineering, including natural polymers, synthetic polymers, multi-component composite polymers, and organic–inorganic polymers, depending on the required properties, as polymer fibers can be designed according to specific needs [23,53]. 

The native ECM is a three-dimensional scaffold consisting of polysaccharide fibers and proteins, found on the nanoscale of tens of nanometers to hundreds of nanometers [5]. Therefore, electrospun fibers can mimic the ECM structure, providing a three-dimensional space and more adhesion sites for cell growth [10]. Based on these techniques, the scaffolds have well-designed shapes and highly improved properties. However, the adjustment of the morphological characteristics in the preparation of the nerve scaffold must consider the parameters of the polymeric solution properties (concentration, molecular weight, solvent used, and type of polymer); of the polymer solution (viscosity, surface tension, and conductivity), the process parameters (voltage, flow rate, collector type, distance between the tip and the collector), and the environmental parameters [54,55].

### 4.2. Quantitative Characteristics

#### 4.2.1. Scaffold Wall Thickness

The tubular NGC dimensions could influence the quality of nerve regeneration and maturation [37]. Some studies have shown that greater wall thickness and reduced porosity have a detrimental effect on axon growth due to the low infiltration and concentration of nutrients and oxygen below 2.0 μg/mL [35,36]. Axon growth was significantly reduced in conduits with wall thicknesses greater than 810 µm (Figure 2c) [36]. High stiffness could be the factor for duct failure with increased wall thickness. 

Balanced strength and flexibility are required for the neural scaffold to withstand the pressure of manipulation, suturing, and surrounding tissue, while a rigid material could damage nerve stumps or surrounding tissues [23,25]. In addition, the thickness of the duct wall must be such that it allows sufficient diffusion of nutrients [25,35,43]; therefore, a suitable wall should provide sufficient mechanical strength with minimum thickness [33,44,47]. 

The optimal wall thickness for implantation was suggested to be 250 μm [23]. Lower wall thickness may be useful for surgical maneuvers [56], but mechanical failures could occur. This was observed in an in vivo experiment, where a thickness of 100 and 200 μm leads to collapse and resistance of less than 0.02 N/mm (Figure 2a) [47]. 

Therefore, a wall thickness ranging from 250 to 400 μm could provide a comparable resistance to commercial NGCs [10,47], thus, favoring the development of thicker axons and normal distribution of blood vessels, providing a thin wall that allows little swelling and without causing nerve compression (Figure 2b) [37].

An increase in wall thickness and decrease in porosity produced greater retention of growth factors within the NGC lumen, which improved the survival capacity of neurons [35,36]. However, these conditions led to a decrease in the amount of oxygen and exchange of nutrients such as glucose and lysozyme through the walls between the internal and external environment [44,47] and favored the formation of fibrous tissue, and more pronounced inflammation of the NGC during degradation, which caused occlusion of the lumen of the nerve guide, graphically represented in Figure 2c [37].

#### 4.2.2. Permeability

##### Porosity of the Scaffold

The percentage of porosity is the main parameter that determines both the diffusion of hydrophilic proteins and the permeability of molecules such as glucose [44], oxygen, nutrients, and neurotrophic factors (NTFs) through the scaffold wall [4,8,13], which stimulates and promotes the orientation of Schwann cells (SCs), fibroblasts, and axonal growth after a lesion in a peripheral nerve [8,12,13,44,45,47,48,49,52].

The incorporation of an optimal range of porosity of 60–80% [36,44,48,49] significantly favored nerve regeneration and promoted the increase in the permeability of nutrients necessary for axonal repair [44,45,48], even better than an autograft in vivo [49]. A lower porosity rate became almost impermeable for molecules such as glucose or lysozymes (Figure 3) [44].

Adequate porosity (60–80%) favored the mechanical properties of NGC with results close to those of native peripheral nerves [48], providing appropriate flexibility that allowed movement at the injury site without twisting [45]. Previous studies revealed that the mechanical properties were inversely proportional to the pore size, and as the pore size increased, the mechanical properties and the elastic limit decreased [45,48]. Porosity over 80% caused mechanical instability (Figure 3), so providing a balance between the number of pores and their size is of vital importance in tissue engineering, to provide adequate mechanical properties and an optimal degradation rate [44,48]. 

##### Pore Size

The superiority of conduits with “asymmetric” or “directional” porosity, i.e., the outer surface of the NGC with a pore size greater than the surface of the NGC lumen, has been widely described in the literature as preventing fibrous tissue infiltration while permeating nutrients, oxygen, and growth factors [24]. In vitro [43] and in vivo [12,51] studies using an asymmetric NGC with pores of 50 nm–10 μm on the inner surface revealed an improvement in nerve regeneration due to the minimization of fibrous tissue infiltration (10 μm) and the escape of NTF [12,43,47,50,51]. Furthermore, an asymmetric NGC with small pores on the inner surface facilitated rapid drainage of exudates from nerve wounds [51], while providing high permeability of nutrients (4–12 nm) and oxygen [12,51], which enabled the development of abundant myelinated nerve fibers of small diameter (Figure 4) [50]. The incorporation of nerve growth factor (NGF) in these scaffolds favored longitudinal axonal growth, forming a bridge between the proximal and distal stumps at 4 weeks, with greater axon density and diameter [51]. 

NGC with internal micro-sized pores of 10–230 µm promoted vascularization within the NCG [45]. However, it favored the invasion of fibrous tissue throughout the lumen, creating an environment less conducive to axonal growth, resulting in a low density and number of nerve fibers and slower axonal growth [50]. This indicated that nerve regeneration through NGC with the microporous inner surface was prevented, even with NGF incorporated [51]. In an asymmetric NGC, the outer surface with a micrometric pore (50 μm) allowed vascular growth for an effective supply of nutrients and oxygen [12,43].

This directional porosity of the NGC, with pores of 50 nm–10 μm on the inner surface and pores of 50 μm on the outer surface, made possible rapid and continuous axonal growth in the proximal and the distal direction in approximately 4 weeks. In addition, it led to a larger axonal diameter and myelin sheath, and faster nerve conduction velocity than non-porous silicone tubes [12], showing a significantly faster recovery from muscle atrophy due to better reinnervation of the muscle compared to NGC with internal micro-sized pores [51]. However, these parameters were low compared to the normal nerve [12].

The degradation rate of NGCs is directly proportional to pore size and porosity [48,57], where a large pore size will favor their degradation. Vijayavenkataraman et al. [48] demonstrated almost 3% more weight loss in scaffolds with a pore size of 550 µm at 28 days compared to scaffolds with a pore size of 125 µm. The faster rate of scaffold hydrolysis with larger pores is due to autocatalyzed degradation [48]. An asymmetric pore distribution with a nanometric pore on the inner surface and a micrometric pore on the outer surface with a porosity greater than 60% allows the impregnation of nutrients but avoids cellular infiltration, which could be a promising strategy to accelerate peripheral nerve regeneration, since, under these conditions, it provides the desirable properties of an ideal NGC structure [46,48].

#### 4.2.3. Diameter of Electrospun Fiber

The diameter of the electrospun fiber can vary according to the properties of the polymer such as concentration, viscosity, and molecular weight [38,43]. Different studies have reported that fiber diameter can alter cell morphology, proliferation, and migration [8,10]. However, it is not clear whether nano- or micro- fibers better support nerve regeneration since optimal results have been reported in vitro using both nanofibers [5,38] and microfibers [8,28,29,41,42], and in vivo evaluation of the effect of the size of the fiber on nerve regeneration is still limited.

A notable impact of filament diameter on neurite growth pattern [8,28], Schwann cell migration rate, degree of neurite unidirectionality, and axon density [28] has been demonstrated. These effects became more prominent in subcellular size ranges or cellular size than in supracellular diameter [28]. Within the range of subcellular or cellular, neuronal cells exhibited greater growth, alignment, and differentiation with fibers of 1.3 µm [29], 5 µm [41], 13.5 µm [8] and 17 µm [43] to 30 µm [28] than with fibers smaller than 0.2 µm [29].

It has been suggested that when the filament size is in the cellular or subcellular size range, growth cones can easily detect the energy differences of different growth directions; however, when the filament size is extremely small, for example, much thinner than the physiological size of the axon, growth cones may not be able to sense energy differences. Thus, the nanometric diameter of fibers of 293 ± 65 nm had a negative influence on the growth of neurites and the migration of SCs, showing a decreased length of neurites compared to fibers of an intermediate diameter (759 ± 179 nm) and large diameter (1325 ± 383 nm) after 5 days of dorsal root ganglion culture [29]. Thus, since the 1.3 µm fibers were more densely packed than with intermediate and small diameter fibers, these packed fibers not only acted as a guiding cue to direct neurite extension and SC migration but also acted as a barrier to impede neurites and SCs from crossing onto nearby fibers [29]. However, when the fibers were of supracellular size (500 µm), the neurites were organized into more densely packed fascicles that impeded adequate axonal growth [28,42]. Therefore, packing between fibers is another parameter that affects neurite extension, and SC migration [29] can be favorable to a degree that provides an available surface that is optimal for cells to adhere and grow [28].

Co-cultures of neuronal and Schwann cells, or dorsal root ganglion (DRG) cultures in polymeric fibers of 1 μm showed superior performance in neurite growth, migration, and SC elongation compared to larger diameter fibers (8 μm) [41]. Two studies corroborate these results [41], obtaining better regenerative results in scaffolds assembled with fibers of 1 µm [29,42]. In the first study, axonal growth, cell migration rate, and higher axonal density were reported in cells cultured in microfibers of 1.0 to 1.3 μm [42]. In the second, the nanometric diameter of the fibers had a negative influence on the growth of neurites and the migration of SCs, since the fibers with a diameter of 293 ± 65 nm showed a decreased length of neurites compared to the fibers of intermediate diameter (759 ± 179 nm) and large diameter (1325 ± 383 nm) after 5 days of DRG culture [29]. 

The alignment of fibers with a small diameter (293 + 65 nm) was statistically different from those with a larger diameter (759 + 179 nm or 1325 + 383 nm). The inability of small fibers to create highly aligned fibers could decrease neurite and SC migration. Because neurites and SCs are much larger than the small-diameter fibers, they may have a more difficult time detecting individual fibers and deciding which groups of fibers to migrate along [29].

By contrast, some studies reported better results with nanofibrous scaffolds [5,38,40,42]. The 300–600 nm nanofibers promoted the organization of the actin cytoskeleton, increasing the adhesion and proliferation rate of SCs [42]. Furthermore, a higher neural stem cell (NSC) differentiation rate was reported, as well as better neurite growth in nanofiber scaffolds (300 nm) than for microfibers (1.5 μm). These results were independent of the alignment of the fibers; however, the diameter did not show a significant effect on cell orientation [38]. 

Another study also revealed better results in nanofibers (251 ± 32 nm) compared to microfibers (981 ± 83 nm), in this case, nanometric fibers resulted in more myelinated axons and thicker myelin sheaths [5]. The G-ratio obtained with NGC from nanofibers was similar to the autograft. Furthermore, the nanofibers appeared to promote better functional recovery with higher compound muscle action potential (CMAP) amplitude and distal motor latency values than the microfiber nerve canals [5]. And finally, another study reported no significant differences in the length of neurites in aligned fibers of different diameters in ranges from 0.8 to 8.8 μm; however, the diameter of the neurite was significantly greater in aligned fibers than in random ones [39].

These results reveal that the diameter of the fiber influences nerve regeneration. However, there is still no consensus regarding the most appropriate fiber diameters for the construction of an NGC using the electrospinning technique. However, according to the results obtained in these studies, most positive results were reported for nerve regeneration in the range of 300 nm to 30 µm.

### 4.3. Methods and Processing Parameters to Produce NGC

#### 4.3.1. Electrospinning

Ideal NGC should be able to mimic the natural environment of the ECM [5,10,53]. An alternative that can emulate the ECM is electrospinning [5,8,10,11]. Electrospinning is a versatile manufacturing method used to produce random or aligned fibers with nanoscale or microscale diameters [8,10]. Based on this technique, the morphological characteristics of the NGC will be influenced by the properties of the polymer solution, the process parameters, and the environmental parameters [54,55]. Studies have used a high voltage in the range of 12 kV to 18 kV [8,29,38,39,41] to generate a polymer jet to form the polymeric fiber [38]. A collector rotating at 1000 [38,39] to 2000 RPM [8] and a distance collection from 9 to 10 cm [8,38,39] was used to obtain an optimal polymeric fiber.

The results indicated that the fiber diameter was directly proportional to the polymer concentration [8,38]. Yang, et al. [38] increased the PLLA concentration from 2% to 5% and the fiber diameter increased significantly from 300 nm to 1.5 μm [38]. On the other hand, it was assumed that the diameter of the fiber was directly proportional to the flow rate (mL/h) and the collection distance (cm) was inversely proportional to the speed of rotation of the collector (RPM) [8,39,41].

The scaffold wall thickness can be manufactured to any thickness by adjusting the collection time, but it was found that the fiber orientation became disordered in the upper layer of the electrospinning sheet when the collection time was greater than 30 min, which may have been due to residual loads in the collecting fibers [39].

#### 4.3.2. Immersion–Precipitation Method and Immersion–Precipitation Phase Inversion Using a Casting Process

The superiority of porous NGC has been amply demonstrated in the literature. Different methods of producing porous scaffolds have been identified in this review: electrospinning [4,47], immersion–precipitation method [13,49,51,52], NaCl as a porosifying agent, [44,45,50], modified immersion precipitation [12,43], freeze-drying and freeze-cast molding method [46] and electrohydrodynamic jet 3D printing (EHD-jetting) [48]. It has been previously described [24] that asymmetric-porosity interconnected pores, that is, the external surface of the NGC with a pore size greater than the surface of NGC lumen, significantly favors the nerve regeneration. 

To obtain this asymmetric pore distribution, studies have used the immersion precipitation method [12,43,51,52], or immersion–precipitation phase inversion using a casting process [13,49]. For the immersion precipitation method, the polymer is dissolved in tatraglycol [12,43,51,52] and then Pluronic F127 powder [12,43,51,52] is added. The asymmetrically porous membrane is obtained after washing with excess water to remove residual solvent and then vacuum-drying [43,51,52]. For the immersion–precipitation phase inversion, after being dipped in a PLGA solution, the glass mandrels were dipped in isopropyl alcohol. The glass mandrels were repeatedly washed in deionized water [13,49]. Asymmetric conduit porosity decreased as the concentration of isopropyl alcohol increased [13].

#### 4.3.3. NaCl Used as a Porosifying Agent

Porosity was introduced with the addition of NaCl as a porogen. To achieve particles with a mean diameter of 17–20 μm, the salt was ground via ball milling to reduce the particle size [44,45,50]. In this NGC fabrication method, the percentage of porosity varied altering the volume percentages of sodium chloride in the polymer–solvent suspension [44].

#### 4.3.4. Freeze-Drying and Freeze-Casting Methods

The freeze-drying method creates randomly oriented pores and comprises four main steps: freezing, ice sublimation, desorption of water bound to the solid, and a packing process to avoid absorption of water from the atmosphere [46].

The freeze casting (ice-templating) method is a simple and nature-inspired technique to produce complex-shaped constructs with interconnected pores. In freeze casting, after transferring the solution into a non-conducting mold, freezing occurs under liquid nitrogen. The frozen solvent acts as a binder and prevents the collapse of the structure. The unidirectional pores of the freeze-cast samples cause a considerable increase in absorption, biodegradation rate, mechanical strength, and drug release level [46]. Among all the methods, freeze casting has gained popularity in tissue engineering applications due to its unidirectional pores, improving physicochemical and mechanical properties, and higher simulation of natural tissues [46].

### 4.4. Mechanical Properties of the Scaffolds

Mechanical properties ensure the stability of the scaffold [17]. Therefore, a scaffold must have adequate mechanical resistance to guarantee the space among cells necessary for the subsequent formation of the extracellular matrix and neural extension [3] and allow the tissue movements [44,46].

A variety of different factors, such as the type of material, morphological characteristics such as porosity, size, and pore distribution, affect compressive strength and mechanical integrity [46].

#### 4.4.1. Mechanical Properties and Polymeric Biomaterial

The type of biomaterial used to produce the NGC has been shown to have a significant role in the mechanical properties of the scaffold [43,45]. Natural polymers have decreased mechanical properties and have rapid degradation rates, which limits their exclusive use. The presence of synthetic polymers improves these properties [58]. However, biomaterials classified within the same category of polymers synthetics differ in mechanical properties. The PLGA scaffolds were approximately 30 times stiffer than the PCL with higher deformation [45]. Furthermore, the incorporation of additives such as Pluronic F127 improves the hydrophilicity of NGC, however, there is a decrease in mechanical resistance with an increase in Pluronic F127 composition. Therefore, the selection of the biomaterial in the development of the NGC is decisive in the mechanical properties.

#### 4.4.2. Influence of Morphometric Characteristics on Mechanical Properties

The morphological properties of NGC are also decisive in the mechanical properties [25,45,47,48]. It has been shown that the deformation behavior of the scaffolds depends on the material to a great extent. However, an increase in porosity also increased compliance in an NGC of the same polymeric material (compliance; a measure of how easily the material could be compressed) [45].

As the pore size increased, the percentage of porosity increased accordingly [48]. The mechanical properties were inversely proportional to the pore size and consequently to the scaffold porosity [48]. The Young’s modulus, yield stress, yield strain, ultimate stress, and ultimate strain were influenced by pore size. The Young’s modulus of the NGC structure decreased from 275 ± 13 to 121 ± 16 MPa with an increasing pore size from 125 to 550 µm respectively. Similarly, by increasing the size pore from 125 to 550 µm the yield stress also had a decrease from 24 ± 3 to 5.6 ± 2 MPa respectively. The ultimate strength of the structure decreased from 32 ± 2.4 to 9 ± 1.4 MPa with increasing size from 125 to 550 µm. Furthermore, the percentage decrease in the mechanical properties as a function of time was greater in scaffolds with a larger pore size and was the smallest in scaffolds with a smaller pore size [48].

Similarly, the NGC wall thickness directly influenced this mechanical resistance, which is important to maintain a stable support structure for nerve regeneration [46]. A wall thickness of smaller than 100 µm collapses without additional force. A thickness of 200 µm provides less strength than 0.02 N/mm [47]. Increasing the thickness of the NGC wall increases the resistance of the channel to resist compression [47]. However, thicker NGC walls interfere in the exchange of molecules [16]. Therefore, a wall thickness within the optimal range of 250–400 µm provides resistance of 0.05 to 0.065 N/mm comparable to commercially available nerve canals, while facilitating the exchange of molecules [47].

### 4.5. Biodegradable NGC Properties

Biodegradation is one of the key factors in determining NGC efficiency [46]. Gradual degradation is required according to axonal growth rates to provide mechanical support during nerve regeneration [47]. The morphological characteristics were decisive in the biodegradation of NGC. Biodegradability is inversely proportional to fiber diameter [52] and is directly proportional to porosity and pore size [52]. The maximum value of weight loss observed was 3.89%, which occured in PCL scaffolds with a pore size of 550 μm on day 28 and scaffolds with a pore size of 125 μm that had a weight 1.38% of their initial weight on day 28 [48]. Over time the fibers loosened and the walls thinned [48]. Therefore, it is important to consider the morphological characteristics so that the NGC maintains its shape and protects the regenerating tissue until functional recovery is achieved.

### 4.6. NGC Hydrophilicity

Surface hydrophilicity in NGC plays an important role in effective nutrient permeability, as well as greater adhesion and initial cell proliferation to the surface [4,12,46]. Hydrophilicity is desirable for NGC because body fluid, including nutrients and oxygen, can be supplied sufficiently in hydrophilized porous NGC [43]. This is why different additives have been incorporated to improve hydrophilicity such as Pluronic F127 [12,43,51] or gelatin [46].

Various studies have shown that morphological characteristics of NGC influence hydrophilicity [4,10]. The hydrophilicity of aligned nanofibers is higher than that of random nanofibers [4,10]. The increased wettability can be attributed to increased porosity and the needle-like pore shape of the aligned nanofibers. Thus, the aligned nanopattern surface can be utilized to fabricate polymeric biomedical scaffolds with strong wettability properties [4]. This characteristic suggested that the hydrophilicity of the nanofibrous mats was influenced by the surface chemical properties and their topography [10]. For a hydrophobic material, the percentage of porosity is the main effect that determines both the diffusion of hydrophilic proteins and the permeability of small molecules [44].

## 5. Conclusions

For successful nerve regeneration, a bioartificial nerve graft must be designed taking into account the materials used, as well as the parameters that affect the local environment. The results of the present review provide basic information on the morphology of the NCG to support the regeneration of peripheral nerves. For the adjustment of those morphological characteristics for the development of the nerve scaffold, the parameters of the polymeric solution must be considered.

The studies selected identified that porosity, pore size, and wall thickness are characteristics that play a fundamental role in the exchange of oxygen, nutrients, and neurotrophic factors, necessary for axonal repair and axonal growth, after nerve injury.

Improved nerve regeneration was demonstrated by a porous scaffold of 60–80% of porosity with asymmetric pore distribution; with nanometric pore size (50 nm–10 μm) on the inner surface and macrometric pores (50 μm) on the outer surface. This also allowed greater permeability for outflow rather than inflow and better SC proliferation and inhibition of fibroblast division. Additionally, a wall thickness ranging from 250 and 400 μm provided sufficient mechanical strength with a minimum thickness to allow the exchange of molecules. Another determining characteristic in nerve regeneration was the size of the polymeric fiber; however a method to obtain optimal results in nano and microfibers was controversial in the literature. These morphometric characteristics influence the mechanical properties, biodegradation, and hydrophilicity of NGC. The incorporation of these parameters for the characteristics as described in the literature will help to reduce the number of in vitro and in vivo experimental studies since it will provide the initial morphological parameters for the development of future scaffolds.

## Figures and Tables

**Figure 1 polymers-14-00397-f001:**
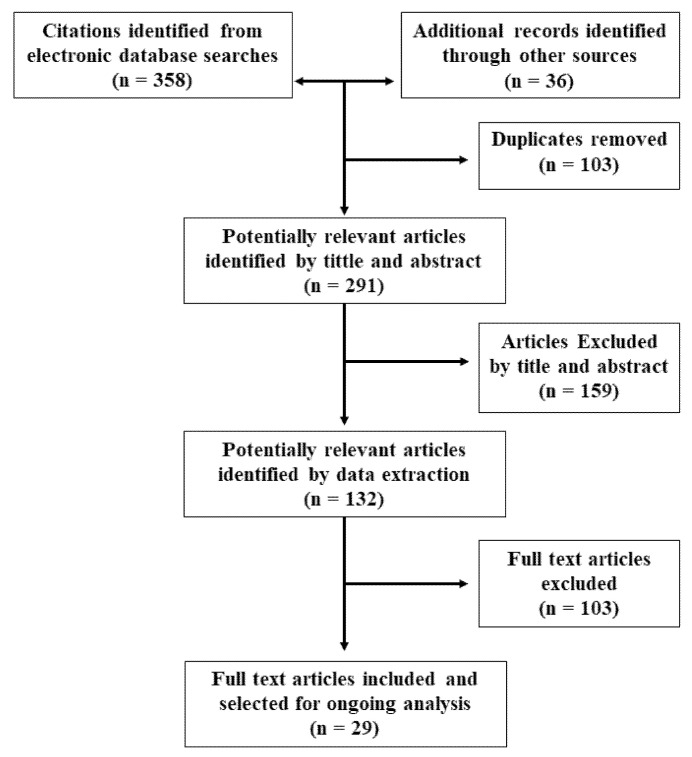
Flow chart for study selection.

**Figure 2 polymers-14-00397-f002:**
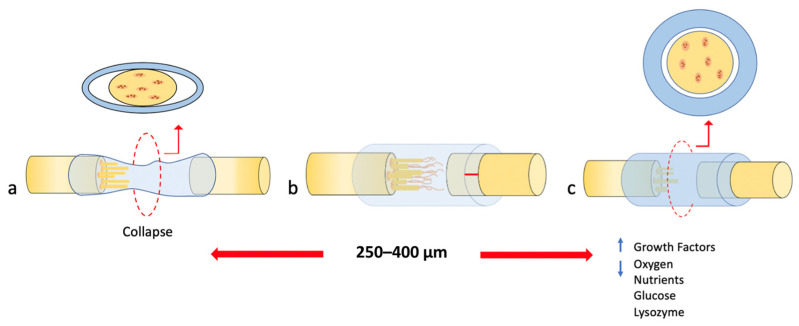
Extracted and modified from Part I of this study [24]. The behavior of the different wall thicknesses in the collapse of the NGC. (**a**) Shows a wall thickness lower than 250–400 μm resulting in the collapse of the NGC. (**b**) Shows the optimal wall thickness in a range of 250–400 μm resulting in successful nerve regeneration. (**c**) Shows a wall thickness greater than 250–400 μm resulting in greater retention of growth factors within the lumen, but decreased oxygen for and exchange of nutrients such as glucose and lysozyme through the walls.

**Figure 3 polymers-14-00397-f003:**
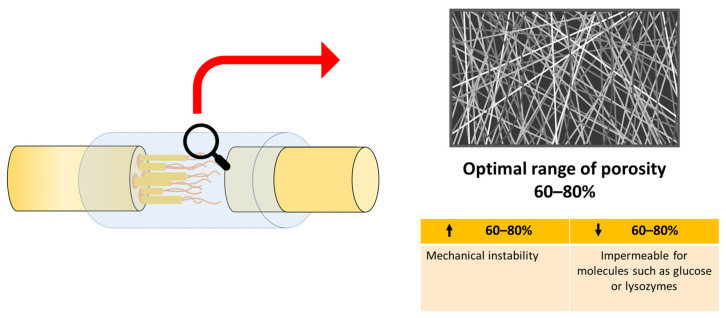
Extracted and modified from Part I of this study [24]. Schematic diagram of optimal porosity. Lower porosity (↓ 60–80%) provides impermeable for molecules. Higher porosity (↑ 60–80%)) provides mechanically instability.

**Figure 4 polymers-14-00397-f004:**
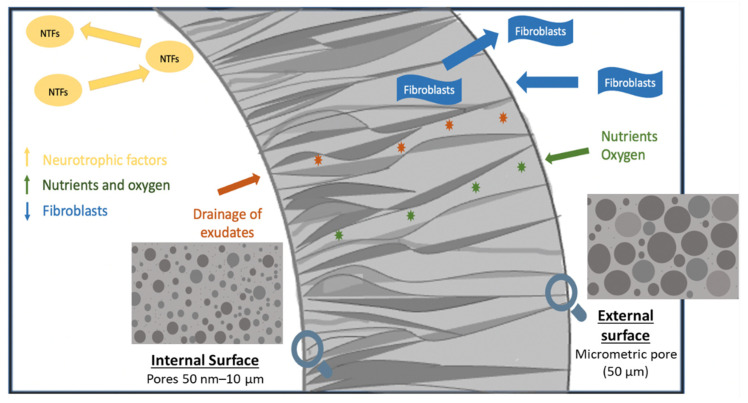
Graphic summary of the asymmetric NGC with pores 50 nm–10 μm on the inner surface and pores of 50 μm on the outer surface. This distribution of pore sizes improved nerve regeneration by minimizing the infiltration of fibrous tissue and the escape of neurotrophic factors (NTFs). In addition, asymmetric NGC facilitated the rapid drainage of exudates (orange stars) from nerve wounds, while providing high permeability of nutrients and oxygen (green stars).

**Table 1 polymers-14-00397-t001:** Studies evaluating the NGC wall thicknesses.

Study	Type of Study	NGC Material	Quantitative Parameters	Main Outcome
Rutkowski et al., 2008 [35]	In vitro (In silico): Schwann cell cultures. Dorsal root ganglia (DRG) of Sprague.	Bioartificial nerve graft (BNG) composed of a tubular conduit of poly-D, L-lactide	Computer model predicting the wall thickness, porosity, and Schwann cell seeding density needed to maximize the axon extension rate and ensuring sufficient nutrients to the neurons.	Low porosity, greater wall and Schwann cell layer thickness led to a decrease in the amount of oxygen available to the axons and greater NGF concentrations.
Rutkowski et al., 2008 [36]	In vitro: SC Cultures. DRG of Sprague–Dawley rats.	Bioartificial nerve graft (BNG) composed of a tubular conduit of poly-D, L-lactide with Schwann cells	Porosities: 0.55–0.95 Wall thickness: 0.56–1.47 mm	Reduced axon growth in conduits with wall thicknesses greater than 0.81 mm, a greater wall thickness, and lower porosities have a detrimental effect on the growth of the axons.
Mobasseri et al., 2015 [25]	In vitro: stem cells differentiated to Schwann cell-like cells. In vivo: Sprague–Dawley rat sciatic nerve injury (*n* = 24).	Poly ε-caprolactone (PCL) and polylactic acid (PLA)	Wall thickness: 70, 100, 120, 210 µm	Increasing the wall thickness also increases stiffness and limits the permeability of the canal, so it did not show any positive effect on the biological response of the regenerating nerve.
Pateman et al., 2015 [23]	In vitro: SC and DRG. In vivo: common fibular nerve of mouse (*n* = 18).	poly(ethylene glycol) (PEG)	Wall thickness: 250 μm	NGC with 1 mm (internal diameter), 5 mm (long) and 250 μm wall thickness supported reinnervation through a 3 mm lesion space after 21 days, similar results to an autograft control.
Den Dunnen et al., 1995 [37]	In vivo: Sciatic nerve of rat (*n* = 24).	50% lactic acid (LA) and 50% e-caprolactone (CL)	Type 1: N° of dip-coated: 2, Int. diameter: 1.23 mm Wall thickness: 0.34 mm. Type 2: N° of dip-coated: 3, Int. diameter:1.18 mm Wall thickness: 0.43 mm. Type 3: N° of dip-coated: 4, Int. diameter: 1.15 mm Wall thickness: 0.64 mm. Type 4: N° of dip-coated: 5 Int. diameter: 1.12 mm Wall thickness: 0.68 mm.	Better nerve regeneration with Type 1 (large amount of targeted neural tissue, a minimal amount of fibrous or scar tissue, and a normal amount and distribution of blood). Type 2: more fibrous tissue and had less favorable nerve regeneration due to nerve compression. Types 3 and 4: bone exposed by severe self-mutilation. Due to the swelling, the NGC lumen completely disappeared.
Den Dunnen et al., 1998 [33]	In vivo: Sciatic nerve of rat (*n* = 30).	Copolymer of DL-lactide and e-caprolactone	Type 1: N° of dip-coated: 2, Int. diameter: 1.23 mm Wall thickness: 0.34 mm. Type 2: N° of dip-coated:3, Int. diameter:1.18 mm Wall thickness: 0.43 mm. Type 3: N° of dip-coated:4, Int. diameter: 1.15 mm Wall thickness: 0.64 mm. Type 4: N° of dip-coated: 5 Int. diameter: 1.12 mm Wall thickness: 0.68 mm.	Thicker NGC wall: swelling of the degrading biomaterial will be so severe that the NGC becomes occluded. Thinner NGC wall: the nerve guides collapsed. Peripheral nerve regeneration across a 10 mm nerve gap inside a P(DLLA-e-CL) nerve guide was faster and qualitatively better in comparison with a 7 mm long autologous nerve graft.

**Table 2 polymers-14-00397-t002:** Studies evaluating the NGC diameter of polymeric fibers.

Study	Type of Study	NGC Material	Technique	Diameter of Fibers	Main Outcome
Wen X. et al., 2005 [28]	In vitro: DRG explants	Poly(acrylonitrile-co-vinyl chloride) (PAN-PVC)	Wet-phase inversion process	Diameters: 5 ± 1.58, 30 ± 1.26, 100 ± 1.41, 200 ± 2.03, 500 ± 3.66 μm	5 and 30 μm-diameter filaments increase the neurite outgrowth and SC migration. Robust and uniformly distributed neuronal growth was achieved with highly directional filaments.
Yang F. et al., 2005 [38]	In vitro: Neural stem cells (NSCs)	Poly(L-lactic acid) (PLLA)	Electrospinning	Aligned fibers: 300 nm nanometer scale 1.5 μm submicron scale	No significant changes to cell orientation were associated with fiber diameters. NSC differentiation was higher for PLLA nanofibers than microfibers, independent of their alignment. The main growing direction of NSC neurites was parallel to nano and microfibers.
Yao L.et al., 2009 [39]	In vitro: PC12 cells	Poly(caprolactone) (PCL)	Electrospinning	Randomly oriented fibers: 4.4 ± 0.5 μm. Aligned oriented fibers: 0.8 ± 0.7, 3.7 ± 0.5, 5.9 ± 0.9, 8.8 ± 0.9 μm.	PC12 cells’ neurites showed similar parallel growth on the aligned fibers irrespective of fiber diameter. Neurite length on aligned fibers (fiber Φ: 3.7 ± 0.5 and 5.9 ± 0.9 μm), was longer than neurite length on randomly oriented fibers.
Wang HB. et al., 2010 [29]	In vitro: Dorsal root ganglia (DRG)	Poly-L-lactic acid (PLLA)	Electrospinning	Highly aligned, electro-spun fiber scaffolds, fiber diameters: Large:1325 + 383 nm interm.:759 + 179 nm small: 293 + 65 nm	Small diameter: did not promote extensive neurite extension or SC migration. Intermediate diameter: promoted long, directed neurite extension independent of SC migration. Large diameter: promoted long, directed neurite extension and SC migration.
Junxia Wang, et al., 2012 [40]	In vitro: human embryonic stem cells (hESCs)	Tussah silk fibroin (TSF)	Electrospinning	Different diameter: 400 and 800 nm	Neurite outgrowth along the fibers was longer on aligned 400 nm TSF-scaffold than on-aligned 800 nm TSF-scaffold. 400 nm aligned TSF scaffold supports survival and promotes neuronal differentiation of hESC-derived NPs.
Daud M.F.B., et al., 2012 [41]	In vitro: I. neuronal or primary SC cultures; II. Neuronal and primary SC in co-culture; III. Isolated DRG cultures, containing both neuronal and SC.	Poly(caprolactone)	Electrospinning	Diameters: 1, 5 and 8 μm	For neuronal cells alone, 8 μm fibers promoted better neurite outgrowth. For neuronal cells plus primary SC or DRG explants, 1 μm fibers supported superior neurite outgrowth, SC migration, and elongation in comparison with 5 and 8 μm fibers.
Gnavi S. et al., 2014 [42]	In vitro: Explant cultures of SC and DRG Ex vivo: SC	Gelatin	Electrospinning	Nanofibrous matrices, diameters: 300 or 600 nm, 1000 or 1300 nm	Nanofibers (300 nm) promoted cell spreading and actin cytoskeleton organization, increasing cellular adhesion and SC proliferation rate. Migration rate and motility, axonal density was greater in cells cultured on microfibers (1300 nm). Microfibers promoted SC migration and axonal outgrowth. Nanofibers promoted SC proliferation and adhesion.
Hu J. et al., 2016 [10]	In vitro: PC12 cells	Poly(ε-caprolactone) (PCL)-Nerve Growth Factor (NGF) and Bovine Serum Albumin (BSA)	Emulsion electrospinning technique	Diameter Random: 343 ± 113 nm; 320 ± 97 nm; 371 ± 95 nm; 343 ± 113 nm Aligned: 354 ± 91 nm; 302 ± 70 nm; 333 ± 90 nm; 320 ± 87 nm.	Aligned nanofibers presented similar diameters to randomly aligned nanofibers, but the aligned nanofibers were more uniform. PC12 neurite length on PCL-NGF and BSA scaffold (diameter 320 ± 87) was higher on aligned nanofibers (70.17 μm) compared to randomly aligned nanofibers (41.67 μm).
Lizarraga LR. et al., 2019 [8]	In vitro: NG108-15 neuronal cells and Schwann cells	Poly(3-hydroxybutyrate) P(3HB) poly(3-hydroxyoctanoate) P(3HO) 25:75 % P(3HO)/P(3HB) blend (PHA blend)	Electrospinning	Highly aligned and uniform fibers diameters: Small: 2.4 ± 0.3 μm Medium: 3.7 ± 0.3μm Large: 13.5 ± 2.3 μm	A direct correlation between fiber diameter and neuronal growth and differentiation was noted. Highly aligned large fibers (13.50 ± 2.33 μm) resulted in better neurite outgrowth and higher neuronal cell differentiation in co-culture. With RN22 SC, the number of NG108-15 cells increased as the fiber diameter increased.
Jiang et al., 2012 [5]	In vivo: sciatic nerve injury model in female Sprague–Dawley rats (*n* = 26).	Poly(ε-caprolactone) (PCL)	Electrospinning	Microfibers (981 ± 83 nm) and nanofibers (251 ± 32 nm)	Nanofiber NGC resulted in a higher number of myelinated axons, thicker myelin sheaths, an increase in regenerated DRG sensory neurons, and functional recovery compared to microfiber and film NGC. Nanofiber conduits possessed a smaller pore size compared to microfiber conduits.

**Table 3 polymers-14-00397-t003:** Studies evaluating NGC porosity, and size of pores.

Study	Type of Study	Material	Technique	Porosity, Size or Distribution of Pores	Main Outcome
Oh et al., 2007 [43]	Pre-experimental study of biomaterials development	Poly(lactic-co-glycolic acid) PLGA and Pluronic F127	Modified immersion precipitation	Inner tube surface: nano-pores ~50 nm/Outer tube surface: micropores ~50 μm	PLGA/F127 tube (3 wt%): optimal mechanical properties and hydrophilicity. Highly effective for nutrient permeability. The tubes show a decrease in mechanical resistance with an increase in the Pluronic F127 compositions.
Kokai et al., 2009 [44]	Pre-experimental study of biomaterials development	Poly(caprolactone) (PCL)	Dip-coating/ salt-leaching technique	Wall thickness: 0.2, 0.6 mm Porosities: 50, 80% Pore size:10–38; 75–150 μm	NGC (0.6 mm) decreased lysozyme loss (~10%) without diminishing glucose permeability. Low porosity NGC (50% porous) showed smooth inner walls and several blind-ended or closed pores. High porosity NGC (80%) showed fewer smooth walls with highly interconnected through-pores for transluminal flow and solute diffusion. NGC (0.6 mm; 10–38 μm pores, 50% porous) were almost impermeable for glucose and lysozyme.
Pawelec et al., 2019 [45]	Pre-experimental study of biomaterials development	Poly(lactide co-glycolide) (PLGA) Poly(caprolactone) (PCL)	Polymer and salt slurry	Relative density of porous films 70 vol% porosity and non-porous films Wall thickness: 61.5–150 µm	Porosity in the scaffold increased compliance from 0.05 ± 0.1 in non-porous PCL to 1.75 ± 0.2 in porous PCL. Porosity decreased flexural stiffness (×10^−2^ N / mm) from 57.40 ± 16.0 in non-porous PCL to 0.88 ± 0.4 in non-porous. In addition, the porous PLGA scaffolds were approximately 30 times stiffer than the porous PCL with higher deformation. On the other hand, the deformation behavior of the scaffolds depended to a great extent on the material. Porous PCL scaffolds exhibited less than 30% permanent deformation after compression. In contrast, the porous PLGA scaffolds experienced a deformation of more than 45%.
Kim et al., 2016 [4]	In vitro: PC12 and S42 cells	Poly lactic-co-glycolic acid (PLGA) and polyurethane (PU)	Electrospinning	Highly-aligned nanofibers and randomly-oriented nanofibers on a single mat with nano to micro sized pores (50 nm–5 μm)	The average diameter of the pores in the aligned nanofibrous mat is three times larger than that in the randomly-oriented mat. The porosity of the aligned nanofibrous scaffolds was higher. Aligned nanofibers served as a guide for neural cells and were able to achieve a higher cell proliferation and migration compared to randomly oriented nanofibers.
Ghorbani et al., 2017 [46]	In vitro: L929 fibroblast cells	Poly (lactic-*co*-glycolic acid) (PLGA)	Freeze-drying and freeze-cast molding method	Porosity (%): 96.33 or 96.16 Pore size (μm): 111.32 ± 160.2; 138.93 ± 302.6 and 152.71 ± 679.9	Randomly oriented pore (freeze-dried) and interconnected pore (freeze-cast) NGC stimulate ECM to support cellular adhesion and migration. Different NGC manufacturing processes affect their properties by altering the microstructure of pores.
Huang et al., 2018 [47]	In vitro: DRG cells cultures	Poly(ε-caprolactone) (PCL) sheaths and collagen-chitosan (O-CCH) filler.	Electrospinning	Pores size: 6.5 ± 3.3 μm Wall thickness: 100, 200, 400 μm	NGC (100 µm) collapsed without additional force. NGC (200 µm) provided a strength lower than 0.02 N/mm at a lateral displacement of 0.3 mm. NGC (400 µm) provided a strength of 0.05–0.065 N/mm at a lateral displacement of 0.3 mm, comparable to commercially available NGC. A PCL porous sheath (pore size: 6.52 ± 3.28 μm) prevented fibroblast invasion and provided mechanical strength for fixation and resistance to compression, exhibiting the appropriate porosity to ensure the supply of oxygen and nutrients, also preventing fibrous tissue infiltration.
Vijayavenkataraman et al., 2018 [48]	In vitro: PC12 cells	Poly(ε-caprolactone) (PCL)	Electrohydrodynamic jet 3D printing (EHD-jetting)	Different pore sizes scaffolds (125–550 μm) and porosities (65–88%).	The Young’s modulus of the NGC structure decreases with increasing pore size from 275 ± 13 to 121 ± 16 MPa. Similarly, the yield stress also has a decreasing trend with increasing pore size from 24 ± 3 to 5.6 ± 2 MPa. The ultimate strength of the structure decreases from 32 ± 2.4 to 9 ± 1.4 MPa. Desirable NGC structure was observed to have 125 ± 15 μm pores. Porosity over 60%: Mechanical properties closer to the native peripheral nerves, and an optimal degradation rate in nerve regeneration post-injury. The percentage decrease of the mechanical properties from day 0 to day 28 was greater in the scaffolds with a greater pore size (550 μm) (~30 to 66%) and was the least in scaffolds with a smaller pore size (125 μm) (~22–45%).
Chan et al., 2007 [13]	In vitro: SC and fibroblasts In vivo: Sciatic nerve of Sprague–Dawley rats	Poly(DL-lactic acid-*co*-glycolic acid) (PLGA)	Immersion–precipitation phase inversion using a casting process	Asymmetric conduits with: high-porosity (permeability) 83.5 ± 5.3%; Medium-porosity (high outflow and low inflow) 73.6 ± 4.7 %; Low-porosity (permeability) 66.1 ± 3.4%.	NGC with different porosities prevented fibrous scar tissue invasion. Allowing the permeation of nutrients, oxygen, and proliferation of SC. Patent directional NGC showed more type A and B myelin fibers in the middle duct and distal nerve compared to the high bidirectional patency NGC.
Chang et al., 2006 [49]	In vivo: sciatic nerve defects in Sprague–Dawley rats (*n* = 80).	Poly(DL-lactic acid-*co*-glycolic acid) (PLGA)	Immersion–precipitation phase inversion using a casting process	NGC: Asymmetric: macrovoids (outer layer), and interconnected micropores (inner layer), possessed characters of larger outflow rate than inflow rate. Autografts Silicone Non-asymmetric	Asymmetric PLGA NGC showed a stable supporting structure, inhibiting exogenous cell invasion during the regeneration process, higher regenerated axons at the mid-conduit, and distal nerve site of implanted grafts compared to the silicone and non-asymmetric groups at 4 and 6 weeks. The asymmetric structure in the conduit wall enhanced the removal of the blockage of the waste drain from the inner inflamed wound in the early stage.
Vleggeert-Lankamp et al., 2006 [50]	In vivo: sciatic nerve of female Wistar rat (*n* = 38).	Poly(ε-caprolactone)	NaCl used as a porosifying agent in the preparation of porous structures	Autografted; grafted nonporous; grafted with pores: outer layer: macroporous (10–230 μm) and inner layer microporous (1–10 μm), macroporous (10–230 μm) or nonporous.	Microporous nerve grafts performed better than nonporous and macroporous grafts. Formation of a tissue bridge with a large diameter, myelinated nerve fibers, more nerve fibers present distal to the graft, the electrophysiological response rate was higher, and the decrease in muscle cross-sectional area was smaller.
Oh et al., 2008 [12]	In vivo: Sciatic nerve of Sprague–Dawley rats (*n* = 63).	Poly(lactic-co-glycolic acid) (PLGA) and Pluronic F127	Modified immersion precipitation method	Porosity: inner surface of the tube with nano-size pores (~50 nm); outer surface with micro-size pores (~50 μm) Nonporous: silicon tubes	PLGA/Pluronic F127 NGC (inner surface pore: ~50 nm) prevented the infiltration of fibrous tissue, retained neurotrophic factors, and provided optimal nutrient infiltration. NGC with the outer surface with micro-sized pores (~50 μm) allowed vascular growth for effective delivery of nutrients and oxygen, allowing rapid and continuous axonal growth from the proximal to the distal direction in ~4 weeks.
Oh et al., 2012 [51]	In vivo: Sciatic nerve of rats (*n* = 48).	Poly(caprolactone) (PCL)/Pluronic F127	Immersion precipitation method	Membrane with nano-size pores (~100 nm) and opposite surface (mold contact side) with micro-size pores (~200 μm)	Nerve fibers regenerated along the longitudinal direction through the NGC with a nano-porous inner surface, while they were grown toward the porous wall of the NGC with a micro-porous inner surface.
Choi et al., 2014 [52]	In vivo: Recurrent laryngeal nerve of female New Zealand rabbits (*n* = 28).	Poly(caprolactone) (PCL)/Pluronic F127	Immersion precipitation method	Asymmetrically porous NGC with selective permeability (inner surface, nano-sized pores; outer surface, micro-sized pores) Nonporous silicone tube. Wall thickness ~0.4 mm, inner diameter of ~1.5 mm and a length of ~12 mm.	Significantly better vocal cord paralysis in the asymmetrically porous PCL/F127 NGC than in the silicone tube. Asymmetrically porous PCL/F127 NGC tubes facilitated nerve regeneration compared with nonporous silicone tubes.

## Data Availability

The data presented in this study are available on request from the corresponding author.

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
