# Peer review of "Optimal Morphometric Characteristics of a Tubular Polymeric Scaffold to Promote Peripheral Nerve Regeneration: A Scoping Review"

_polymers, 2022, doi:10.3390/polym14030397_

Round 1

Reviewer 1 Report

The present review tackles a paramount topic in the field of biomaterials for optimal applications to improve human health, i.e. the peripheral nerve regeneration. Development of suitable scaffolds is a well known requirement that must favour  stem cells growth  and bio-clinical issues.

The review adopts the standard criteria for a useful review, addressed to cover selected 4 morphometric properties for promoting nerve regeneration: wall thickness, fiber size, pore size, and porosity.

In my opinion, the review is complete and at the same time concise, references are selected accurately and relevant papers are selected and discussed; their number is limited to be handled by an interested reader.

Specifically, the discussions on selected electrospinning  approaches are very useful and of high relevance for developments.

Presentation is very clear and well-organized.

In conclusions, I would recommend the presentation of the present scope review.

Author Response

The authors are deeply grateful for their analysis and comments on the manuscript.

Reviewer 2 Report

This review article focuses on the influence of morphometric parameters (e.g. wall thickness, fiber size, pore size, and porosity) of NGC on the supportive environment characteristic (e.g. nutrient permeability, retention of neurotrophic factors, and optimal mechanical Properties) for nerve regeneration application. The article used three tables to summarize wall thickness, fiber size, pore size, and porosity on the performance of the NGC. The article missed the followings:

  • Discuss briefly methods and processing parameters (e.g. electrospinning, near-field electrostatic printing) to produce the scaffold.
  • Revise each table based on in vitro and in vivo studies.
  • Add mechanical properties values for each NGC scaffold studied in this article instead of mentioning one scaffold is better than other with respect to mechanical properties (e.g. stiffer, high compliance)
  • Some major outcomes such electrical conductivity, degradation, thermal stablity ignored during the study results and discussion.

Author Response

Discuss briefly methods and processing parameters (e.g. electrospinning, near-field electrostatic printing) to produce the scaffold.

Reply: Thank you for this comment. We briefly discussed the methods to produce the scaffolds used in the selected studies and their processing parameters as suggested.

Revise each table based on in vitro and in vivo studies.

Reply: The authors appreciate the comment. The three study tables were revised and reorganized, so that they are initiated by pre-experimental studies, followed by in vitro studies, and finally by in vivo studies, informing in the latter case which animal species and nerves were studied. Finally, we included information on the total number of animals used.

Add mechanical properties values for each NGC scaffold studied in this article instead of mentioning one scaffold is better than other with respect to mechanical properties (e.g. stiffer, high compliance)

Reply: Thank you for this comment. In the studies that analyzed the mechanical properties of NGC, their values ​​were added to the tables. In addition, a paragraph about the influence of morphological characteristics and materials on mechanical properties was included in the discussion.

Some major outcomes such electrical conductivity, degradation, thermal stability ignored during the study results and discussion.

Reply: Thank you for this comment. We include in the discussion the influence of morphological characteristics on the biodegradation and hydrophilicity of NGC. However, the other characteristics suggested to be included were not significantly described in the selected studies of the present review.

Round 2

Reviewer 2 Report

The authors took care my comments.